# Hierarchical organization of urban mobility and its connection with city livability

Aleix Bassolas [1], Hugo Barbosa-Filho [2], Brian Dickinson[3], Xerxes Dotiwalla[4], Paul Eastham[4], Riccardo Gallotti [5], Gourab Ghoshal[2,6]*, Bryant Gipson[4], Surendra A. Hazarie[2], Henry Kautz[3,6], Onur Kucuktunc[4], Allison Lieber[4], Adam Sadilek [4] & José J. Ramasco [1]*

The recent trend of rapid urbanization makes it imperative to understand urban characteristics such as infrastructure, population distribution, jobs, and services that play a key role in urban livability and sustainability. A healthy debate exists on what constitutes optimal structure regarding livability in cities, interpolating, for instance, between mono- and polycentric organization. Here anonymous and aggregated flows generated from three hundred million users, opted-in to Location History, are used to extract global Intra-urban trips. We develop a metric that allows us to classify cities and to establish a connection between mobility organization and key urban indicators. We demonstrate that cities with strong hierarchical mobility structure display an extensive use of public transport, higher levels of walkability, lower pollutant emissions per capita and better health indicators. Our framework outperforms previous metrics, is highly scalable and can be deployed with little cost, even in areas without resources for traditional data collection.

[1] Instituto de Física Interdisciplinar y Sistemas Complejos IFISC (CSIC-UIB), Campus UIB, 07122 Palma de Mallorca, Spain. [2] Department of Physics & Astronomy, University of Rochester, Rochester, NY 14627, USA. [3] Department of Computer Science, University of Rochester, Rochester, NY 14627, USA. [4] Google Inc., 1600 Amphitheatre Parkway, Mountain View, CA 94043, USA. [5] Bruno Kessler Foundation (FBK), 38123 Trento, Italy. [6] Goergen Institute for Data Science, University of Rochester, Rochester, NY 14627, USA. *email: gghoshal@pas.rochester.edu; jramasco@ifisc.uib-csic.es

Rapid urbanization has led to over half of the world's population living in cities, making it crucial to understand urban characteristics such as infrastructure, facilities, population distribution, jobs, and services that play a key role in health, urban livability, and sustainability[1,2]. The recent availability of digital traces from Information and Communications Technologies has enabled the study of urban systems with unprecedented spatiotemporal resolution, giving rise to a highly interdisciplinary endeavor, loosely termed as the *science of cities*[3–5]. Past research indicates a connection between urban characteristics and city spatial organization, giving rise to a debate regarding the optimality of such structures as it relates to livability[6,7]. It has been argued that the increase of population, along with the congestion induced by the concentration of activity, drives cities from a monocentric to a polycentric configuration[8,9]. While polycentricity accounts for the number of distinct activity centers, urban sprawl is related to how spatially scattered they are. Cities can be compact, with all centers in a single district, or sprawled if the centers are far apart, although typically, they exist in a continuum between these two limits[10].

Indicators of urban organization have been proposed based on population density, land use, employment, and infrastructure distributions[11–16]. While, traditionally, these metrics are estimated from surveys and satellite imagery[17], emergent mobile technologies allow for a more direct, timely, and precise measurement of human mobility[18–23]. Mobility, as a product of the citizens needs, residence/job locations and transportation infrastructures, bears important insights on the dynamics taking place in cities[10,24–26]. Empirically, it has been observed that the concentration of jobs, amenities, services and other related socioeconomic activity is distributed across multiple spatial centers—so-called hotspots—the geography of which strongly influences mobility flows[26,27]. Given this connection, the location of these hotspots can be directly extracted from mobility[28,29].

Here, we use anonymous and aggregated flows from users who opted-in to Location History to extract patterns of global intraurban trips to quantify the organization of urban mobility. We begin by extracting the hierarchical structure of hotspots in cities, and develop a metric, the flow-hierarchy, that is based on the interaction between hotspots. The metric allows us to classify cities based on their level of dynamical hierarchy and thus establish a connection to key urban indicators. Specifically, we show that cities with larger mobility hierarchy display more population-mixing, extensive use of public transportation, higher levels of walkability, lower pollutant emissions per capita and better health indicators. The flow-hierarchy contains more information on urban features than traditionally used metrics such as population density and urban sprawl. We conclude with a discussion of the implications of our findings, including possible policy directions as it relates to urban planning.

## Results

**Aggregation of Trip Flows.** The anonymous mobility patterns are aggregated over Google users who opted-in to Location History [https://support.google.com/accounts/answer/3118687]. This dataset contains trip flow information from over 300 million people world-wide (including residents and visitors of cities) for the year 2016, aggregated weekly and with spatial granularity corresponding to S2 cells [https://github.com/google/s2geometry] of approximately 1.27 km$^2$ (Details in Supplementary Note 1). These trip flows were obtained within the framework of the Mobility Map project, in which machine learning techniques were applied to anonymized logs data to segment a raw GPS trace into semantic trips. The system automatically found trips by taking

into account a variety of signals, such as timing of location points, dwell times, and other factors[30]. All trips were anonymized and aggregated by jointly applying differential privacy via the Laplace mechanism in combination with k-anonymity. The automated Laplace mechanism adds random noise drawn from a zero mean Laplace distribution and yields $(\epsilon, \delta)$-differential privacy guarantee of $\epsilon = 0.66$ and $\delta = 2.1 \times 10^{-29}$, which is very strong. The parameter $\epsilon$ controls the noise intensity in terms of its variance, while $\delta$ represents the deviation from pure $\epsilon$-privacy. The closer it is to zero, the stronger the privacy guarantees. For example, with these values of the parameters, an attacker knowing that the dataset was generated using $\epsilon = 0.66$ and desiring to know whether a user was included would at best improve the level of certainty over a random guess by approximately 16%. Further information on the aggregation procedure can be found at[31] and [https://policies.google.com/technologies/anonymization]. No individual user data was ever manually inspected, only heavily aggregated flows of large populations were handled.

The trip flow data can be interpreted as weighted networks formed by S2 cells as nodes, and flows as link weights. Figure 1a shows the network in North America and the inset focuses on the New York City area (intra-cell flows are not included in the analysis). In what is to follow, we analyze 127 American cities (those with a population greater than 400,000), and 174 of the most populated global cities that are present in our dataset (see Supplementary Note 2, Supplementary Figs. 1 and 2 and Supplementary Tables 1 and 3 for details). The network sizes are roughly the same order of magnitude as those that can be constructed from commuting data available in the US census (although our data contains much more mobility information than merely commuting). For example, New York City, the most populated US metropolis with almost 20 million inhabitants, has a network of 6,213 cells and 110,798 connections (an average degree of 17.8). Medium-sized cities such as Atlanta with a population around 5 million has a mobility flow network with 4156 cells and 46,333 links. (See Supplementary Table 2 for a list of US cities with the number of cells and links).

**Hotspots and flow-hierarchy.** Hotspots are identified by setting a threshold on the number of outgoing trips in every cell (incoming and outgoing trips are roughly symmetric (see Supplementary Figs. 3 and 4). The threshold for each hotspot level is assigned by iteratively applying a nonparametric method based on the derivative of the Lorenz curve[26]. The Lorenz curve is the sorted cumulative distribution of outflows and is obtained by plotting, in ascending order, the normalized cumulative number of nodes vs. the fraction of total outflow. The threshold is then obtained by taking the derivative of the Lorenz curve at (1, 1) and extrapolating it to the point at which it intersects the x-axis. Once hotspot nodes at a level $\ell$ have been extracted, they are excluded from the distribution, and the threshold is recalculated such that new hotspots at level $\ell + 1$ are assigned (see Supplementary Note 3 and Supplementary Fig. 5 for more details on the method). The procedure is illustrated up to level $\ell = 5$ for New York City in Fig. 1b, where Lorenz curves are depicted in progressively transparent shades of blue (and corresponding slopes from red to yellow) as one goes down in levels.

The extracted spatial distribution of hotspot levels display a range of patterns across cities; examples are shown in Fig. 1c–e for Paris, Bangkok and Los Angeles (~10 million inhabitants) and in Fig. 1f–h for Alexandria, Santiago de Chile, and Sydney (~5 million). For those cities with similar populations (rows in the Figure), the observed spatial distribution is substantially different: Paris and Alexandria have hotspot levels organized in an onion-

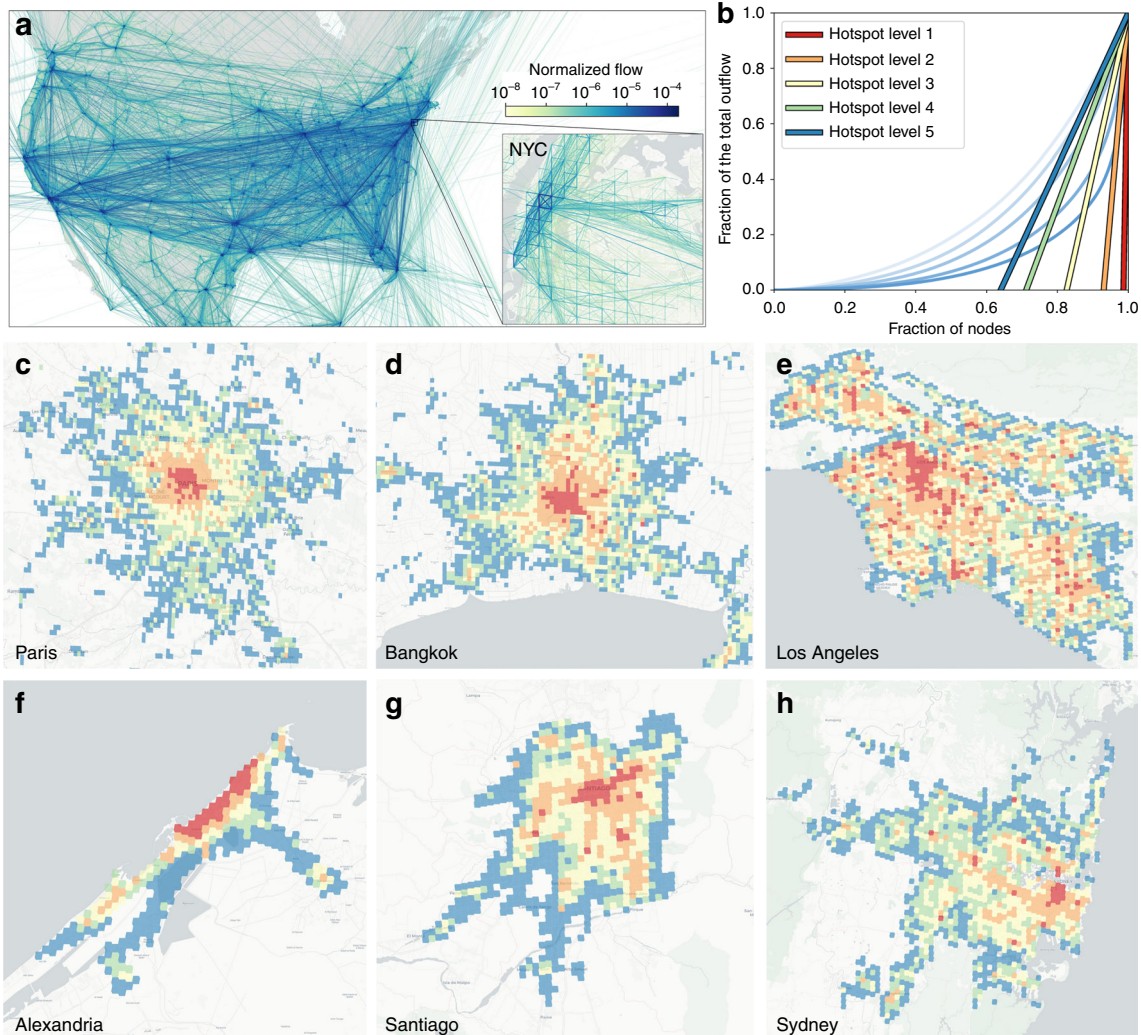

**Fig. 1** Human mobility and hierarchical structure of cities. **a** Mobility network extracted from North America (New York City shown in inset). Nodes are geographical units (S2 cells) and links are weighted by flows between locations with darker colors corresponding to more intense flows. **b** Hotspot level calculation using the Loubar method. **c-h** Maps of hierarchical hotspots for two groups of three metropolitan areas with similar population: **c** Paris (France) (12.4 million inhabitants), **d** Bangkok (Thailand) (14.5 million), **e** Los Angeles (USA) (13.35 million inhabitants), **f** Alexandria (Egypt) (5.17 million inhabitants), **g** Santiago (Chile) (7.11 million), and **h** Sydney (Australia) (5.13 million inhabitants). The color code is the same as in panel **b**: level 1 (dark red), level 2 (orange), level 3 (yellow), level 4 (green), and level 5 and below (dark blue). The underground map layout is produced using Carto. Map tiles by Carto, under CC BY 3.0. Data by OpenStreetMap, under ODbL

like structure; Bangkok and Santiago have hotspots that are spread out away from the center; and, finally, those in Los Angeles and Sydney are scattered across the city. Apart from the spatial distribution, the number of levels as well as the number of hotspots per-level vary across cities (see Supplementary Fig. 5c, d). However, the corresponding distribution of total outflow is relatively stable across cities and is fairly heterogeneous, with the majority of flows contained in the first three or four levels (Supplementary Fig. 5e, f).

The top-heavy nature of the flows, combined with the observed differences in the spatial distribution of hotspots, indicates different degrees of hierarchical structure in terms of mobility. To quantify this hierarchy, we investigate the extent of interaction between hotspots of varying levels of activity. We construct a matrix **T** with elements $T_{ij}$ corresponding to trips between hotspots at levels $i$ and $j$, normalized by the total number of intracity trips, in order to compare across cities with different populations. Since the archetypal hierarchical structure is a tree, the extent to which cities are tree-like or flat in terms of trip flows

is captured by the flow-hierarchy $\Phi$, defined as the tri-diagonal trace of $T$ (see Fig. 2a)

$$\Phi = \sum_{i=1}^{L-1} \left( T_{ii} + T_{i(i+1)} + T_{(i+1)i} \right) + T_{LL}$$
$$= \sum_{i,j=1}^{L} T_{ij}\left( \delta_{ij} + \delta_{i(j-1)} + \delta_{(i-1)j} \right).$$
(1)

Here, $L$ is the total number of hotspot levels (varying from city-to-city) and $\delta_{ij}$ is the Kronecker delta. The metric is one if the flow interaction occurs only between same- or adjacent-level hotspots (tree-like structure), and close to zero if flows are flat, i.e., distributed uniformly across all $L$ hotspot levels. Thus, higher values of $\Phi$ correspond to a stronger hierarchical organization. (See Supplementary Note 4 and Supplementary Figs. 6–8, for details of the calculation, including dependence on city size and boundary effects.)

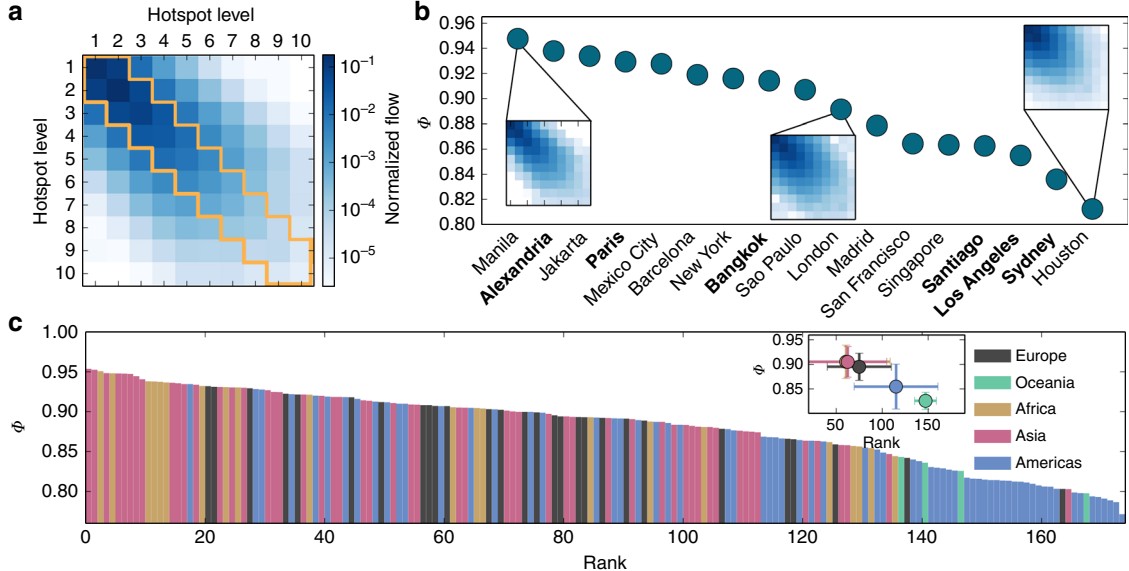

**Fig. 2** City flow-hierarchy $\Phi$ calculation and ranking. **a** Matrix of trip flows $T_{ij}$ between hotspot different levels for New York City (only first ten levels are shown). Each entry of the matrix is normalized by the sum of all flows. The flow-hierarchy $\Phi$ is calculated by summing the tri-diagonal entries marked in yellow. **b** City flow-hierarchy for a subset of ordered cities and the corresponding trip-matrix for three examples: Manila (Philippines), London (UK) and Houston (USA). The cities marked in bold correspond to those shown earlier in Fig. 1c–h indicating a more spatially dispersed layout of the hotspots with decreasing $\Phi$. **c** Full set of global cities ranked according to decreasing values of $\Phi$ and colored according to continent. The inset shows the mean and standard deviations for each continent for both $\Phi$ and the corresponding rank

In Fig. 2b, we show a subset of the cities ranked in terms of the flow-hierarchy along with three examples of the corresponding flow-matrices. Included in the set are the cities whose spatial distribution of hotspots are shown in Fig. 2 (marked in bold). A clear monotonic trend is apparent whereby decreasing values of $\Phi$ correspond to spatially scattered distribution of hotspots, indicating that the flow-hierarchy also contains spatial information of the mobility. The complete set of worldwide cities is shown in Fig. 2c (and Supplementary Table 4), ranked in decreasing order of $\Phi$, and colored according to continent. The figure indicates a remarkable continental trend, whereby Asian and African cities are among the most hierarchical followed by European, American and, finally, those located in Oceania. The trend is even clearer when plotting the average flow-hierarchy per continent against the average ranking (shown as inset).

Given that the definition of $\Phi$ is general and nonparametric, it can be used for any flavor of mobility data[22]. As a comparative exercise, we also calculate $\Phi$ from commuting data for US cities, extracted from the census (Supplementary Note 5 and Supplementary Figs. 10 and 15). As commuting captures only a subset of total mobility, the top level hotspots do not necessarily coincide, although the correspondence is strong in the city center, it is weaker in the suburbs (See Supplementary Fig. 11 for the case of New York City). While commuting captures primarily residential and office locations, the hotspots extracted from the trip-flows also include major transportation hubs, leisure centers, and areas of major economic activity (Supplementary Fig. 12 and Supplementary Table 5). Due to this difference, the flow-hierarchy obtained from commuting data is systematically lower than that obtained from the trip-flows, although notably the trend is the same. In addition, $\Phi$ can be also defined at different spatial scales. Starting from the S2 cells, we have mapped into a square grid of different cell sizes (Supplementary Note 4). We find that the values of $\Phi$ for the American cities is coherent across scales, the value may slightly change but the ranking of cities according to $\Phi$ is stable (Supplementary Fig. 9).

**Null models**. In order to determine whether our calculated values for the flow-hierarchy are a function primarily of the heterogeneous distribution of trips or contain additional information such as the spatial layout of the hotspots, we need a suitable reference value or null model. An obvious first choice would be to consider the case where the movements in cities are fully mixed, or in other words follow an uniform distribution. If flows are distributed uniformly between hotspot levels, then it is easily seen that $\Phi_u = (3L-2)/L^2$. For the case of New York, $L = 14$ and therefore in a variant of the city with uniform flow distribution $\Phi_u = 0.20$, which is significantly lower than the empirically measured value of $\Phi = 0.92$. Given that the observed values in cities lie in a range $0.77 \leq \Phi \leq 0.95$ (Supplementary Table 4), a more robust reference point is needed that takes into account the heterogeneous distribution of flows across hotspot levels (Supplementary Fig. 5). Consequently, we next generate random variants of each city by rewiring the flows across levels while preserving the empirical flow distribution. A null trip matrix can then be produced as

$$T_{ij}^{h} = \sum_{k=1}^{L} T_{ik} \times \frac{\sum_{m=1}^{L} T_{mj}}{\sum_{m,k=1}^{L} T_{mk}}. \qquad (2)$$

Here, $\sum_k T_{ik}$ is the total outflow of level $i$ and $\sum_m T_{mj}/\sum_{mk} T_{mk}$ is the fraction of inflows for level $j$. The resulting flow-hierarchy $\Phi_h$ is a function of the heterogenous distribution of flows and neglects any spatial correlation between the hotspots. In Fig. 3a and b, we show the differences between the randomized and empirical flow matrices for New York City, with the former having a upper-triangular structure in contrast to the nested structure seen in the latter. The resulting value of the flow hierarchy, $\Phi_h = 0.65$, is much lower than the empirical value $\Phi = 0.92$. In Fig. 3c, we show a subset of American cities, comparing them to their randomized counterparts, finding that in all cases the empirical values of $\Phi$ are consistently higher, indicating an

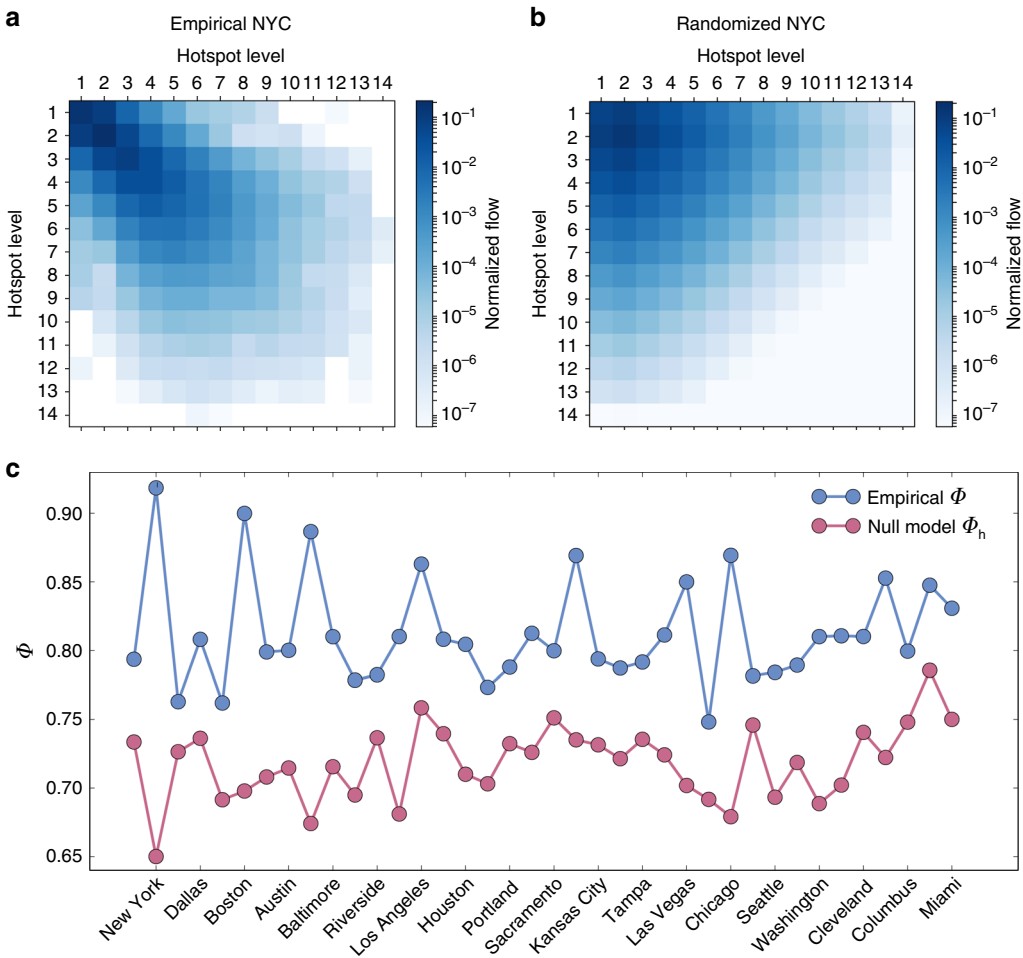

**Fig. 3** Comparison between the observed $\Phi$ and $\Phi_h$ of the heterogeneous random null model. Matrices of flows between hotspots levels for the **a** empirical mobility network of New York City, and **b** the randomized one obtained from Eq. (2). **c** Flow-hierarchy for the randomized network and the corresponding empirical data for a subset of US cities, indicating that real cities are significantly more hierarchical than their randomized versions

inherent hierarchical structure that relates to spatial correlations of the mobility flows, in addition to the heterogeneity in their distribution.

**Connection between urban indicators and flow-hierarchy.**
Next, we examine the connection of the flow-hierarchy with other urban indicators such as transportation, pollutant emissions, and health. We note that connections between such indicators and existing measures of urban structure, such as population density and urban sprawl have been previously established and used to inform urban policy[2,6,16,32–38]. The mobility patterns are ultimately shaped by the urban infrastructure, population distribution and the attendant socioeconomic needs of citizens, therefore it is natural to expect similar connections. Given that $\Phi$ captures the structural organization of mobility across the entire city, it is an ideal metric to study these potential relations. Indeed, cities that are more hierarchical tend to have higher population densities and are more compact (Supplementary Figs. 22, 23, and 25). To investigate the connections, we collected metadata for the full set of US cities, a choice motivated by the fact that the data is relatively homogeneous and reliable, given that it is collected by a single agency (See Supplementary Note 8 for details).

We start by considering the modal transportation share. For cities with an extensive public transportation system, there are likely many transportation hubs, that in principle could facilitate the emergence of hotspots, enhancing the hierarchy of mobility

flows. Thus, a priori, one would expect to see a strong connection between the presence of public transportation (and its use) and stronger hierarchies in the mobility patterns. In Fig. 4a–c, we plot $\Phi$ against various commuting modes: by car, by public transportation and on foot, respectively. The Pearson $R_p^2$, is at least 0.4, with all three $p$ values below 0.001 (full list of $p$ values for all urban indicators in Supplementary Table 6). Strongly hierarchical cities show higher levels of public transportation usage and more pedestrian trips, while transportation in the less hierarchical ones is dominated by the use of private cars. Similar trends exist for other transportation metrics such as vehicle-miles traveled (VMT) per capita and transit trips per capita (Supplementary Fig. 19). We note, however, that while the connection between $\Phi$ and public transportation usage holds on average, the explained variance is 0.45 and there are important exceptions. In particular, both Los Angeles and San Francisco have comparable hierarchical structure, yet differ significantly in their modal share, with the latter displaying a much higher share of public transportation and lower car usage. Indeed, both Los Angeles and San Francisco have comparable values of per-capita VMT (Supplementary Fig. 19a) despite the differences in car modal share, indicating that residents of San Francisco typically travel longer distances than those in Los Angeles. This implies that a centralized public transportation network by itself does not facilitate hierarchy in mobility flows, instead other factors such as spatial constraints, geographic impediments and corresponding land usage are important as well.

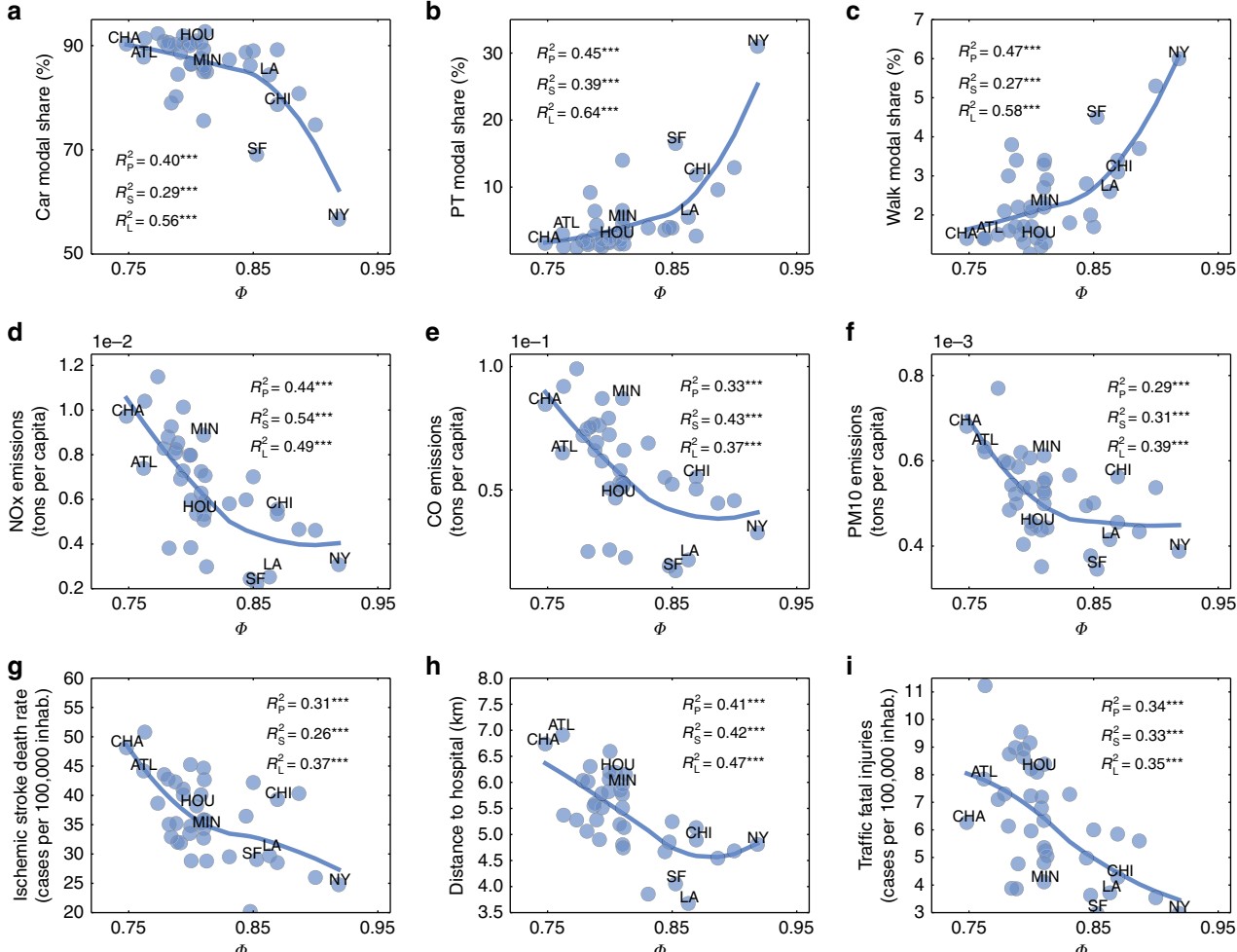

**Fig. 4** Connecting $\Phi$ with urban indicators in US cities. **a–c** Modal share for trips to work (%) as a function of $\Phi$ in US cities: **a** Modal share by private car, **b** by public transportation (PT) and **c** by walking. **d–f** Relation between $\Phi$ and pollutant emissions: **d** NO$_X$, **e** CO and **f** PM10 in metric tons per capita. **g–i** Connecting $\Phi$ and health: **g** ischemic stroke mortality per 100,000 inhabitants, **h** average distance to closest hospital and **i** incidence of traffic fatalities per 100,000 residents. Correlations are measured using Pearson, Spearman, and LOESS with the corresponding explained variances denoted as $R_P^2$, $R_S^2$, and $R_L^2$, (details in Supplementary Note 9). Asterisks correspond to significance-level (p value) of regressions (one * is less than 0.05, two less than 0.01, and three less than 0.001). Some city names appear in the plots: ATL (Atlanta), CHA (Charlotte), CHI (Chicago), HOU (Houston), LA (Los Angeles), MIN (Minneapolis), NY (New York City), and SF (San Francisco)

Given the established connection between car use and pollution[37,38], one might expect to see a similar relation between $\Phi$ and pollution indicators. Yet, once again, the emission of pollutants is not only related to the use of transportation modes. The quality of transportation infrastructure, congestion and geographical features also play an important role. In Fig. 4d–f, we plot $\Phi$ against emission of NO$_X$, CO, and particulate matter (PM10), finding that pollution emissions are anticorrelated with higher levels of hierarchy (See Supplementary Table 7 for connection with PM2.5 emissions). However, the shape of the LOESS fit and the general trend in Fig. 4a are very different from those in Fig. 4d-f (one is concave and the other is convex). Indeed, it appears that the total emissions are shaped more by the per-capita VMT than the car modal share, with similar shapes for the LOESS fit. This is confirmed by the fact that Los Angeles and San Francisco are quite comparable in their pollution emissions and VMT values, despite the large differences in modal share. Given that $\Phi$ also captures the spatial distribution of the hotspots—the distance between which influences the length of trips—if hotspots are organized as in Paris (Fig. 1c) with a single clear nucleus, as

opposed to the more scattered configuration of Los Angeles (Fig. 1e), $\Phi$ is higher and the trips tend to be shorter.

We next explore the relation between $\Phi$ and health indicators on account of the strong observed relation between pollution and public wellness[39,40]. We consider first the prevalence of ischemic stroke, whose incidence is known to be directly affected by the pollutant concentration in the atmosphere[32,36]. While there appears little connection between the flow-hierarchy and prevalence (Supplementary Fig. 21), we find a clear connection with the mortality rate, with a monotonically decreasing trend with increased $\Phi$ (Fig. 4g). It is well-known that the survival rate for strokes is strongly dependent on timely medical attention; either provision of emergency services at home, or ease of access to hospitals[41,42]. Indeed, Fig. 4h shows that the average distance to hospitals from any given area in the city decreases with increasing $\Phi$. This connection is likely due to a combination of factors. First, we note the correlation between $\Phi$ and the average population density as well as its spatial distribution (Supplementary Figs. 22 and 23). Denser cities display a more hierarchical organization of mobility, with a concentration of high-level hotspots in a single area. With the reasonable assumption that

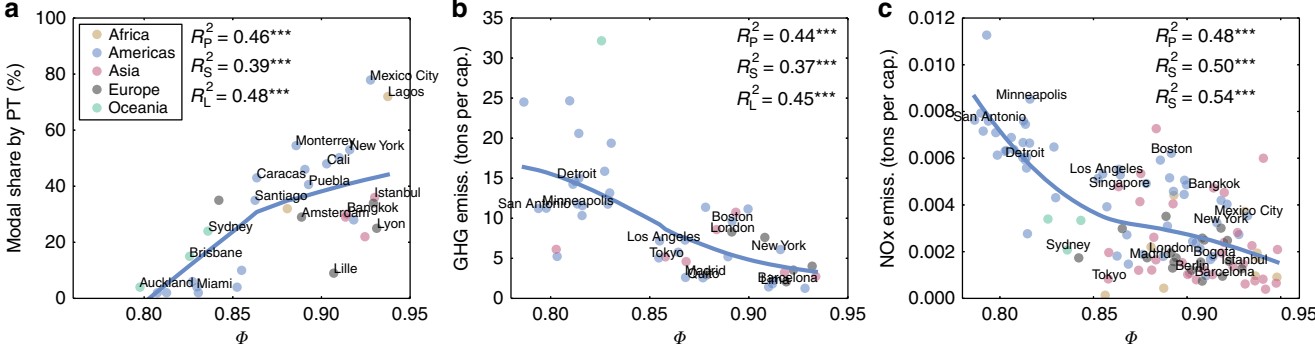

**Fig. 5** Connecting $\Phi$ with urban indicators in worldwide cities. **a–c** Relation between, modal split, pollution and health in worldwide cities. **a** Modal share, **b** greenhouse gas emissions per capita in equivalent metric tons of $CO_2$, **c** $NO_X$ emissions per capita

medical services are distributed according to population, denser cities have comparatively higher number of such facilities in a given area as compared to sparser cities, implying more proximity to such facilities on average. The presence of well developed public transportation, which is more common (on average) in more hierarchical cities, may also play a role as it contributes to decreased traffic congestion and facilitates ease of travel in more dense areas as compared to using cars alone. However, given the comparable mortality and proximity values seen for Los Angeles and San Francisco, which differ in public transportation share, but are similar in terms of per-capita VMT, proximity to medical facilities as a function of the population density and its spatial distribution seem to be the primary causal factors. Note that merely the average population density in itself is not sufficient to explain these trends (Supplementary Fig. 24g–i). The need for prompt medical attention is crucial also in traffic accidents, and we find the same connection between $\Phi$ and accident mortality in Fig. 4i even while accounting for the differences in modal share across cities (Supplementary Fig. 20). The proximity of high-level hotspots in cities with large values of $\Phi$ is also the likely causal mechanism behind the observed greater pedestrian activity, which can contribute to better health.

The results indicate that the flow-hierarchy contains markedly more information on urban indicators than population density (Supplementary Fig. 24) and is comparable to, or in some cases, better than measures of urban sprawl (see Supplementary Table 9, Supplementary Fig. 26). In addition, we find that the correlations are temporally stable (Supplementary Table 8), robust with respect to population thresholds (Supplementary Table 7) as well as to the removal of outlier cities (Supplementary Fig. 18). While it is difficult to extend the complete analysis for global cities—due to limited availability of homogenized data—we can do so for a restricted set of indicators, including transportation and certain types of pollutant emissions. Figure 5a shows the same trend between modes of transportation and $\Phi$ as seen for US cities. In addition, greenhouse-gas (GHG), $NO_X$ and $CO_2$ emissions are anticorrelated with hierarchy (Fig. 5b, c and Supplementary Figs. 27 and 28) in the same fashion as other pollutants seen for US cities.

It is likely that other socioeconomic variables and behavioral factors influence urban indicators, quite apart from the flow-hierarchy. To evaluate such influence, we performed a detailed multivariate analysis considering additional relevant socio-economic variables for each of the studied transportation, environmental and health indicators (Fig. 6). The variables are not necessarily independent or orthogonal to $\Phi$, as can be seen for cases when the blue bar is smaller than the green one. While the inclusion of other variables increases $R^2$, we observe important gains (ranging from 20 to 280%) with the incorporation of $\Phi$. For

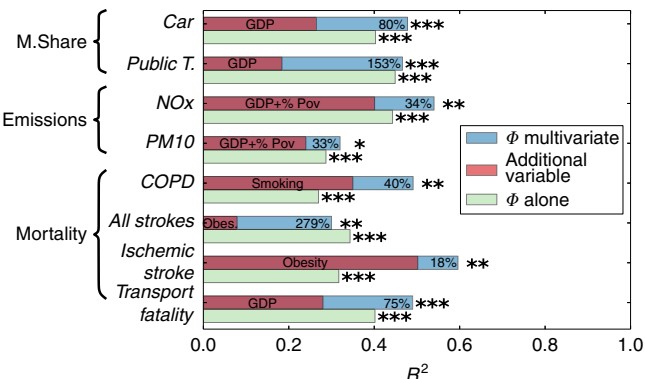

**Fig. 6** Multivariate analysis of mobility share, emissions and mortality causes. The $R^2$ when only $\Phi$ is considered is shown in green. The contribution from at least one or two variables that explain an important part of the variance is shown in red, while in blue we display the additional gain in variance by introducing $\Phi$. The extra variables are GDP (gross domestic product per capita), % Pov (percentage of poverty), smoking (prevalence of smoking in fraction of population) and obesity (prevalence in fraction of population). Data sources are listed in Supplementary Note 8 and the complete set of results for the multivariate analysis are shown in Supplementary Tables 10–30

example, flow-hierarchy explains over 0.3 $R^2$ in chronic obstructive pulmonary disease (COPD) prevalence. Even when smoking—a major factor in COPD—is included in the model, $\Phi$ accounts for an additional 40% of variance (most likely due to is connection to pollution emissions), indicating its ability to capture multiple factors in a single metric. Data sources are listed in Supplementary Note 9, and the full analysis (including a comprehensive set of health and transport indicators) is shown in Supplementary Tables 10–30.

## Discussion

We have presented a thorough investigation of the mobility structure of cities based on how populations traverse high activity areas (hotspots). Our metric, the flow-hierarchy $\Phi$, derived entirely from trip-flows, quantifies the hierarchical organization of urban mobility. While hierarchies across cities have been extensively studied since Christaller's work in the 1930s[43], in terms of one city the focus has been primarily on the differences and comparative advantages between core and peripheral structures, as well as whether cities are mono- or poly-centric. Our results instead show that cities lie across a rich spectrum of hierarchical organization that appears to strongly correlate with a

series of urban indicators including population-mixing (See Supplementary Note 7 and Supplementary Figs. 16 and 17), transportation, pollution, and health. More hierarchical cities (in terms of flow being primarily between hotspots of similar activity levels) tend to have greater levels of mixed-populations, wider use of public transportation, higher levels of walkability, lower pollution emissions, and better indicators of certain measures of health. Where applicable, we have provided hypotheses and plausible explanations for the observed connections between mobility and our considered urban indicators, paving the way for a more thorough investigation into the complex interplay between the considered metrics.

While existing measures of urban structure such as population density and sprawl composite indices correlate with $\Phi$ (Supplementary Figs. 23 and 26) and with urban indicators (Supplementary Figs. 24 and 26), the flow-hierarchy conveys comparatively more information, as also demonstrated through a multivariate analysis that includes behavioral and socioeconomic factors. Furthermore, measures of urban sprawl require composite indices built up from much more detailed information on land use, population, density of jobs, and street geography among others (sometimes up to 20 different variables[6]). In addition to the high data requirements such metrics are also costly to obtain; censuses and surveys require a massive deployment of resources in terms of interviews, and are only standardized at a country level, hindering the correct quantification of sprawl indices at a global scale[44]. On the other hand, the flow-hierarchy, being constructed from mobility information alone, is comparatively much more accessible in terms of both cost and availability of data. Indeed, given the information content inherent in $\Phi$—levels of hierarchy, spatial distribution of activity, quality of urban indicators—it can be deployed efficiently and at scale in those parts of the world where there is little-to-no metadata for urban indicators. The metric is completely general and nonparametric, enabling it to be extracted from any mobility data (including, as we demonstrated, commuting flows extracted from the census), although the most accurate results are obtained when the full spectrum of mobility is considered. Apart from Location History data and surveys, $\Phi$ can easily be calculated from mobile phone records[22] or Location Based Social Networks[23].

Given the ongoing debate on the optimal structure of cities[6,32,37], the flow-hierarchy introduces a different conceptual perspective compared to existing measures, and can shed new light on the organization of cities. Indeed, while the metric is based on trip-flows, mobility, in addition to opportunity and demand, is strongly shaped by geography, land-use and quality of infrastructure. From a public-policy point of view, it is instructive to note that cities with greater degree of mobility hierarchy tend to have more desirable urban indicators. Given that this hierarchy is a measure of proximity and direct connectivity between socioeconomic hubs, a possible direction could be to shape opportunity and demand in a way that facilitates a greater degree of hub-to-hub movement than a hub-to-spoke architecture. The proximity of hubs can be generated through appropriate land use policies, that can be shaped by clever zoning laws in terms of business, residence or service areas. The presence of efficient public transportation and the comparatively lower use of cars is another important factor; although public transportation is not enough in terms of facilitating ease of access and lowering pollution, when countered by comparatively longer car trip-lengths as seen in the case of San Francisco. Perhaps a combination of policies, such as congestion-pricing, used to disincentivize private transportation to socioeconomic hubs, along with building public transportation in a targeted fashion to directly connect the hubs, may well prove useful.

## Methods

**Flow-hierarchy estimation from trip-distribution models.** It is instructive to check whether the flow-hierarchy can be accurately extracted from aggregated trip distribution models such as the gravity, radiation and population-weighted opportunity[45–47]. In principle, these models are expected to generate realistic trip flows between locations (with varying degrees of accuracy), given input information such as the population in the locations or the trip out-flows of every cell.

We focus on US cities and use two trip-flows from the the US census and from our Location History data (Supplementary Note 6). In Supplementary Fig. 13, we show the comparison obtained from the models for each of the inputs. Using only census data as an input (Supplementary Fig. 13a) does a poor job of reproducing the empirical values, but this is to be expected, as the data contains primarily work–home commutes. However, using the trip-outflows from the Location History as input (Supplementary Fig. 13b), we see a much better correlation of $\Phi$ between the models and the data, with the radiation model being the most accurate. Nevertheless, no model exactly reproduces the empirical values; the stronger nonlinearities present in the radiation model foster the flows between top hotspot levels leading to a mild overestimation of $\Phi$, while the linear dependence on the masses in the other two models generates comparatively less flows between high-level hotspots thus leading to an underestimation. This can be seen in the matrices of flows between hotspots (Supplementary Fig. 14 for New York City), where the radiation model produces a hierarchical structure that most closely matches the real structure of New York (Fig. 2a). To confirm this hypothesis, we test a nonlinear version of the gravity model with (Supplementary Fig. 15), finding increasingly good agreement with empirical values as one moves progressively from a linear to quadratic dependence on the masses. In combination, the results indicate that the efficacy of using these models to estimate $\Phi$ varies with th eamount of information available on the different types of mobility. In all cases, however, it appears that the out-flow by itself is a sufficiently good estimator.

## Data availability

In this work, we use the following data sources: the commuting data at the block level was obtained from [https://lehd.ces.census.gov/data]; the modal share of commuting trips for all the US metropolitan areas was obtained from the census as well [https://factfinder.census.gov/faces/nav/jsf/pages/index.xhtml], which provides the percentage of commuting trips in terms of transportation mode. The smoking rate by city was obtained from [https://chronicdata.cdc.gov/Behavioral-Risk-Factors/Behavioral-Risk-Factor-Surveillance-System-BRFSS-P/dttw-5yxu]. Pollutant emissions were obtained from the United States Environmental Protection Agency (US EPA) which makes public a National Emissions Inventory (NEI) every three years. We used the version corresponding to 2014 [https://www.epa.gov/air-emissions-inventories/2014-national-emissions-inventory-nei-data]. The incidence of ischemic stroke and all types of stroke, including morbidity and mortality, has been obtained at the scale of counties was sourced from the Center for Disease Control and Prevention (CDC) of the United States [https://nccd.cdc.gov/dhdspatlas/reports.aspx]. Transport and health indicators related to traffic fatal injuries have been obtained from [https://www.transportation.gov/transportationhealth-tool/indicators]. General mortality and mortality by age were obtained from [http://www.healthdata.org/. Mortality by chronic obstructive pulmonary disease was obtained from [https://ephtracking.cdc.gov/DataExplorer/]. Data about the location of acute care hospitals in the United States was obtained from [https://hifld-geoplatform.opendata.arcgis.com/datasets/a2817bf9632a43f5ad1c6b0c153b0fab_0]. Modal share of transport in worldwide cities was obtained from [https://brtdata.org/]. Greenhouse gas (GHG) emissions were obtained from the Climate Disclosure Project (CDP) [https://www.cdp.net/es]. The OECD also reports $CO_2$ emissions per capita at a world scale within their defined boundaries for a large number of cities [https://measuringurban.oecd.org/]. Emissions of other pollutants globally, including $NO_X$, obtained from the Emissions Database for Global Atmospheric Research [http://edgar.jrc.ec.europa.eu], which provides a grid map of emissions of a set pollutants. The rest of the data may be made available, upon request to the authors.

## Code availability

The code for the analysis was programmed using Python and using standard packages. All the calculations can be reproduced with the equations provided in the main text or the Supplementary Information. Even so, the code used here is available upon request to the authors.

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

## Acknowledgements

We thank Avi Bar, Curt Black, Andrei Broder, Susan Cadrecha, Stephanie Cason, Ciro Cattuto, Charina Chou, Katherine Chou, Iz Conroy, Liz Davidoff, Jeff Dean, Jutta Degener, Damien Desfontaines, Jason Freidenfelds, Evgeniy Gabrilovich, Vivien Hoang, Sarah Holland, Michael Howell, Pan-Pan Jiang, Ali Lange, Bhaskar Mehta, Caitlin Niedermeyer, Genevieve Park, Prem Ramaswami, Chase Rigby, Kathryn Rough, Flavia Sekles, Calvin Seto, Aaron Stein, Chandu Thota, Michele Tizzoni, Alessandro Vespignani, and Ashley Zlatinov for their insights and guidance. A.B. is funded by the Conselleria d'Educacio, Cultura i Universitats of the Government of the Balearic Islands and the European Social Fund. A.B. and J.J.R. also acknowledge partial funding from the Spanish Ministry of Science, Innovation and Universities, the National Agency for Research Funding AEI and FEDER (EU) under the grant PACSS (RTI2018-093732-B-C22) and the Maria de Maeztu program for Units of Excellence in R&D (MDM-2017-0711). G.G. and S.H. acknowledge funding from the Department of Economic Development (DED), New York through the NYS Center of Excellence in Data Science at the University of Rochester (C160189). G.G. and H.B. also acknowledge support in part by the U. S. Army Research Office (ARO) under grant number W911NF-18-1-0421. Any opinions, findings, conclusions or recommendations expressed are those of the author(s) and do not necessarily reflect the views of the DED or the ARO.

## Author contributions

A.B., G.G., H.K., A.S., and J.J.R. developed the concepts and designed the study. A.B. analyzed the data. X.D., P.E., B.G., O.K., A.L., and A.S. computed and provided the mobility map data. A.B., H.B., B.D., R.G., G.G., S.A.H., A.S., and J.J.R. contributed to the work methodology. A.B., R.G., G.G., H.K., A.S., and J.J.R. wrote the paper. G.G., H.K., A.S., and J.J.R. coordinated the study. All authors read, edited, and approved the final version of the paper. Except for the first and last authors, the author names are in alphabetical order.

## Competing interests

The authors declare no competing interests.
