## [Peer Review File · Nature Communications]

Reviewers' comments:

Reviewer #1 (Remarks to the Author):

The paper introduces a new metric related to aggregated mobility flows that is shown to be connected to a number of urban indicators. The results are interesting and the methods are sound.

I suggest the authors to investigate whether the proposed flow hierarchy can be captured by human mobility models. In the paper they show that the uniform flow distribution and the randomized null model are not able to reproduce the observed flow-hierarchy, Φ . I am wondering if it would be possible to obtain a good estimate of Φ using mobility flows generated by state of the art mobility models, like the gravity model, the radiation model and the population weighted opportunity model [Yan, Xiao-Yong, et al. "Universal predictability of mobility patterns in cities." *Journal of The Royal Society Interface* 11.100 (2014): 20140834]. This could enable the possibility to apply the proposed methodology more widely, as the kind of mobility data used in the paper to compute Φ is not publicly available.

Typo: section's title "Hotspots and flow-hierachy" should read "flow-hierarchy".

Reviewer #2 (Remarks to the Author):

The authors assess the hierarchical structure of cities by using large-scale population flow data. They introduce a new metric of hierarchy which is based on a definition of hotspots (urban activity centers) and a subsequent consideration of flows between them. The authors additionally correlate this new metric with various urban measures related to human health.

The paper is clearly written and easy to follow. The general topic of assessing the internal urban structure and dynamics and their relation to the performance of cities is interesting. And the sheer amount of data used is certainly impressive.

However, my main objections are as follows. The new metric seems to be artificial (one could identify hotspots in many different ways; a large fraction of flows in cities is actually between home locations and destinations and back home (see Schneider et al. 2013), and less so between centers as is the focus in the paper). From a policy maker's / urban planner's viewpoint, it is hard to understand how one actually could influence the flow-hierarchy.

Moreover, calculation of the new metric requires massive data but the actual gain in new insights / predictive power compared to more traditional and less data-intensive measures like density/sprawl index is limited. Instead of presenting correlations to health-related issues that are similar to those known from density and sprawl, it would perhaps be more interesting to explore the impact of the flow-hierarchy on the general mixing of people, which is more fundamental for the functioning of cities?

Reviewer #3 (Remarks to the Author):

The authors propose a novel quantitative indicator of hierarchy in urban travel. This is based on stratification of zones in urban areas according to the frequency of travel from (equivalently, to) them to indicate travel hot-spots. Cities in which a greater proportion of travel is between zones that have similar hot-spot levels (ie peer-peer) are described as having a high degree of flow hierarchy according to the authors' proposed measure (Φ), whilst cities in which more travel is between hot-spots and lesser travelled zones are described as having a low degree of flow hierarchy.

The authors show that Φ varies among cities around the world and that it is substantially greater in value than would arise from a null model of origin-destination travel frequency that is proportional to each of origin and destination frequencies (cf contingency table null model).

This description is followed by an investigation of relationships between the proposed Φ measure and various specific urban indicators that include descriptors of travel behaviour (modal shares), environmental measures (pollutant emissions per capita) and public health measures (stroke mortality, traffic fatalities, distance to closest hospital). The authors report weak ($R^{2} \sim 0.26 - 0.64$) but statistically significant correlation across US cities between their novel Φ measure and these measures. They make the point that these relationships are stronger than for established measures of mobility.

The strength of the paper is that it establishes the novel Φ measure of urban mobility that is somewhat outperforms existing ones in correlation with various urban indicators of pollution and health.

There are several weaknesses in the evidence presented and the lines of reasoning based on them. The fundamental point for the authors to not is that the Φ measure of mobility is a descriptor of travel behaviour rather than any of urban form, infrastructure, transport provision and travel opportunity. In this sense, the novel measure is less a description of urban organisation than it is one of citizens' behaviour that represents a response to the combination of opportunity and need. Although the abstract promises relationships between urban organisation and urban indicators, what is actually provided is an observation of the relationships between citizens' travel behaviour and these indicators.

There are more specific weaknesses that arise from the authors' failure to consider causality. The most extreme case of this is to seek a correlation between Φ and distance to closest hospital. No explanation is offered for how either of these could affect the other.

The observation that a the mode share of public transport is greater and that of car is less in cities that have a greater degree of hierarchy risks endogeneity: public transport is necessarily centralised and serves between major trip-making centres. For this reason the causality between Φ and mode share is ephemeral.

The observation that emissions of pollutants tend to be lower in cities that have high Φ values is clearly exposed to common causation.

The present manuscript is observational in nature and lacks hypothesis. The findings are descriptive and offer no guidance on policy. In order to achieve adequate standard for publication the authors need to clarify these matters - and eliminate their references to acausal relationships.

Beyond this, there are several areas where the text requires improvement. These include, but are not necessarily limited to:

- o confusion in the text between "significant" and various of {large, statistical, important, substantial}
- o the term "population flow" is unusual in the transport field: the population itself does not relocate. The terms "travel" or "travel behaviour" might be used instead, at the authors' discretion (and responsibility)
- o The penultimate sentence ("Specifically, we show ...hierarchy." of the introductory section is opaque and requires clarification.
- o The privacy-protecting noise mechanism described in the section "Aggregation of Trip flows" - specifically "This automated mechanism ... very strong guarantee." conveys little information on the processes followed and should therefore be clarified.
- o The symbol Φ is absent from captions to Fig 2 and Fig 4 so should be inserted

- o The caption to Fig 2 refers erroneously to Fig 2c-h (perhaps Fig 1c-h is intended?)
- o The calculation on page 5 of $\Phi_{u} = 0.21$ is not consistent with the formula and parameter values specified.

RESPONSE TO REVIEWERS

Reviewer #1:

The paper introduces a new metric related to aggregated mobility flows that is shown to be connected to a number of urban indicators. The results are interesting and the methods are sound.

We thank the reviewer for the kind assessment and are pleased that our results are found to be of interest.

I suggest the authors to investigate whether the proposed flow hierarchy can be captured by human mobility models. In the paper they show that the uniform flow distribution and the randomized null model are not able to reproduce the observed flow-hierarchy, Φ . I am wondering if it would be possible to obtain a good estimate of Φ using mobility flows generated by state of the art mobility models, like the gravity model, the radiation model and the population weighted opportunity model [Yan, Xiao-Yong, et al. "Universal predictability of mobility patterns in cities." *Journal of The Royal Society Interface* 11.100 (2014): 20140834]. This could enable the possibility to apply the proposed methodology more widely, as the kind of mobility data used in the paper to compute Φ is not publicly available.

This is a very relevant suggestion and we are remiss in not having thought of this ourselves. The idea of using the null model, as the reviewer recognizes, was to determine whether the signal encoded in Φ is robust. The idea of using aggregated mobility models to estimate Φ in situations of data scarcity is indeed an intriguing approach. We note that Φ being a completely general and non-parametric measure, can be used to extract hierarchy from any mobility data. As an example, we added a new analysis and discussion in the main manuscript (**page 6, last para, and Supplementary Note 5, Supplementary Figure 9**) where we computed the hierarchy based on commuting data from the US Census.

Given that the definition of Φ is general and non-parametric it can be used for any flavor of mobility data [22]. As a comparative exercise, we also calculate Φ from commuting data for US cities, extracted from the census (Supplementary Note 5 and Supplementary Figure 9). As commuting captures only a subset of total mobility, the top level hotspots do not necessarily coincide, although the correspondence is strong in the city center, while being weaker in the suburbs (See Supplementary Figs. 10 for the case of New York City). While commuting captures primarily residential and office locations, the hotspots extracted from the trip-flows also include major transportation hubs, leisure centers and areas of major economic activity (Supplementary Figure 11 and Supplementary Table 5). Given this difference, the flow-hierarchy obtained from commuting data is systematically lower than that obtained from the trip-flows, although notably the trend is the same.

We note that the results are quite similar, with a strong correspondence, particularly in the city-center, between the census data and the full mobility spectrum encoded in our Location History Data. The differences are primarily in the suburban areas as commuting flows typically over-represent work-home transits. However, our analysis includes much more than the commuting flows, taking into account the full spectrum of mobility types. Indeed, the hotspots extracted in our analyses include, in addition to densely populated and commercial areas, important transportation infrastructure like subway and bus stops (**see Supplementary Figure S11**). The overall hierarchical trends, however, remain the same. We also note, that the trip-outflows used by us can be estimated (to varying degrees of accuracy) from CDR and Location Based Social Network Data. We added a paragraph on this in the Discussion section (**page 14, first para**):

The metric is completely general and non-parametric, enabling it to be extracted from any mobility data (including, as we demonstrated, commuting flows extracted from the census), although the most accurate results are obtained when the full spectrum of mobility is considered. Apart from Location History data and surveys, Φ can easily be calculated from mobile phone records [22] or Location Based Social Networks [23].

We have also added a new section in the Supplementary Information (**Supplementary Note 6**), where we present an analysis of Φ obtained for US cities, via the three suggested models, as well as a discussion in a new section titled ***Estimating the flow-hierarchy from aggregated trip-distribution models on page 7 of the manuscript:***

It is instructive to check whether the flow-hierarchy can be accurately extracted from aggregated trip distribution models such as the gravity, radiation and population-weighted opportunity [35– 37]. In principle, these models are expected to generate realistic trip flows between locations (with varying degrees of accuracy), given input information such as the population in the locations or the trip out-flows of every cell...

As inputs to the three proposed models, we have employed trip out-flows extracted from the data analyzed in the manuscript (either census commuting or Location History). The findings are quite interesting; using census as an input and comparison basis produces a poor agreement between the estimated and empirical flow-hierarchies (**Supplementary Fig. S12a**). However, using the trip Location History out-flows as input leads to significantly better agreement (**Supplementary Fig. S12b**). Yet, there are differences. While the gravity and PWO models, underestimate Φ , the radiation model leads to a mild overestimation. Indeed, it appears that the empirical trends are well reproduced by the radiation model. This can be understood by how each model determines the hotspots and their inter-connections. The gravity and the PWO models are both linear in the out-flow, implying that hotspots are connected with a much lower flux than in the empirically observed values of Φ . The radiation model being quadratic in the outflow, generates the correct hierarchies, but with a mild overestimation. This can be seen through a non-linear variant of the gravity model, interpolating between linear and quadratic values, with better agreement between the estimate and the empirical values with increasing exponent (**Supplementary Fig. S14**).

In combination, the results indicate that the efficacy of using these models to estimate Φ varies with the amount of information available on the different types of mobility. In all cases, however, it appears that the out-flow by itself is a sufficiently good estimator.

Typo: section's title "Hotspots and flow-hierarchy" should read "flow-hierarchy".

We have corrected this, thanks.

We also point the referee to new analysis on the connection between hierarchy and population-mixing in the **second paragraph of page 8** in the main manuscript where we report a monotonic trend between Φ and the level of mixing.

In summary, we thank the reviewer for prompting us to examine the efficacy of model-estimation. We feel the new results have improved the clarity and reach of the manuscript. We look forward to a favorable appraisal.

Reviewer #2:

The authors assess the hierarchical structure of cities by using large-scale population flow data. They introduce a new metric of hierarchy which is based on a definition of hotspots (urban activity centers) and a subsequent consideration of flows between them. The authors additionally correlate this new metric with various urban measures related to human health.

The paper is clearly written and easy to follow. The general topic of assessing the internal urban structure and dynamics and their relation to the performance of cities is interesting.

We thank the reviewer for the positive assessment and are glad that our topic of research is found to be of interest and significance.

However, my main objections are as follows. The new metric seems to be artificial (one could identify hotspots in many different ways; a large fraction of flows in cities is actually between home locations and destinations and back home (see Schneider et al. 2013), and less so between centers as is the focus in the paper).

We disagree in the assertion that our metric is artificial. Our definition of the flow-hierarchy Φ , is non-parametric, general, and agnostic to the type of data being used, as long as the variable of interest can be classified at different levels. Indeed, it is a rather direct way to quantify the degree of hierarchy in a system. Perhaps, the definition of hotspots is what is being referred to here? In terms of the methodology, the thresholds needed to define a cell as a hotspot in a single iteration of our method might be seen as artificial at first blush, (in much the same way as

setting *any threshold* is artificial), nevertheless the authors of Ref. [26], cited in the manuscript, demonstrate that the Loubar method is a robust way to define a threshold based on the distribution of the variable of interest.

While it is true that the variable that is used to define hotspots can be anything (including the degree, centrality of cells, or even the population), it depends on the context. In this work, we seek to study the patterns of movement between areas of high activity in urban systems, and therefore the trip-flows is the most natural metric.

Furthermore, there appears to be some misunderstanding about the nature of the data. We stress that our data includes *all types of mobility flows* including the work-home commuting trips pointed out by the referee. The hotspots extracted from the wider spectrum of trip-flows correspond to densely populated areas, business districts, leisure centers as well as to transportation hubs (**see Supplementary Figure 11**), exactly as one would expect, and accurately reflect the areas of high activity when one includes all types of mobility data beyond just commuting flows. We emphasize, *there is no trip-selection performed*, the full spectrum of mobility is included. Indeed, while there is a high-frequency of work-home flows, there is no reason why the residential part of these trips will be characterized as hotspots, given that residential neighborhoods are typically dispersed in various parts of the city. The metric quantifies flows *between hotspots*, and, except for very dense zones, residential areas are (in general) not classified as such. Besides, our data includes flows of not just residents, but visitors to the city. Visitors (typically) stay near socioeconomic centers.

Now of course, if one does trip-selection, for example, restricting the analysis to *only commuting flows*, then the kind of behavior pointed out by the referee does show up. We have added a new section where we did the analysis on commuting flows in American cities extracted from the US census, (**see Supplementary Note 5, Supplementary Figs. 9-11 and Supplementary Table 5 as well as the following excerpt on page 6, last para, of the main manuscript**):

Given that the definition of Φ is general and non-parametric it can be used for any flavor of mobility data [22]. As a comparative exercise, we also calculate Φ from commuting data for US cities, extracted from the census (Supplementary Note 5 and Supplementary Figure 9). As commuting captures only a subset of total mobility, the top level hotspots do not necessarily coincide, although the correspondence is strong in the city center, while being weaker in the suburbs (See Supplementary Figs. 10 for the case of New York City). While commuting captures primarily residential and office locations, the hotspots extracted from the trip-flows also include major transportation hubs, leisure centers and areas of major economic activity (Supplementary Figure 11 and Supplementary Table 5). Given this difference, the flow-hierarchy obtained from commuting data is systematically lower than that obtained from the trip-flows, although notably the trend is the same.

We note, that the correspondence between commuting and Location History data is strong in the city-center, with differences appearing in the suburban regions, where residential areas are now classified as higher-level hotspots (in the incomplete dataset), similar in line to what is being suggested by the referee. This is a rather important point, and we believe further buttresses the novelty of our findings. It appears that when one accounts for the full spectrum of mobility inherent in urban centers, the results are somewhat different than what was previously suggested. Indeed, the paper by Schneider *et al* on mobility motifs is an excellent one, however, the study was conducted on two cities, based on survey and CDR data. In contrast, our analysis is at a much higher resolution and at far-greater scale focusing on 127 American cities and 174 global ones, finding consistent trends everywhere. Consequently, we believe our findings are rather robust.

Moreover, calculation of the new metric requires massive data

In fact, it is the exact opposite. As we mentioned, the only input needed to compute the metric are the trip out-flows, in comparison to the noisy, difficult to obtain, and data-intensive sprawl indices. This is an important point that needs clarification, so we have updated the manuscript **on page 3, second para**:

In what is to follow, we analyze 127 American cities (those with a population greater than 400, 000), and 174 of the most populated global cities that are present in our dataset (See Supplementary Note 2, Supplementary Figs. 1-2 and Supplementary Tables 1 and 3 for details). The network sizes are roughly the same order of magnitude as those that can be constructed from commuting data available in the US census (although our data contains much more mobility information than merely commuting). For example, New York City, the most populated US metropolis with almost 20 million inhabitants, has a network of 6213 cells and 110, 798 connections (an average

degree of 17.8). Medium-sized cities such as Atlanta with a population around 5 million has a mobility flow network with 4156 cells and 46, 333 links. (See Supplementary Table 2 for a list of US cities with the number of cells and links.)

Our data consists of aggregated flows between areas of the cities, and not individual trajectories. The information needed for our analysis is of the same order of magnitude per month as the OD matrices extracted from the census. For example, NYC, the largest US metropolis with almost 20 million inhabitants, has a network of 6213 cells and 110798 connections (which means an average in- or out-degree of 17.8). Medium-sized cities such as Atlanta with a population around 5 million has a network consisting of 4156 cells and 46333 links. A table with the number of cells and links has been included in the SI for all the cities in the study (**Supplementary Table 2**). Most of them have less than 1000 cells and an average out-degree around 10 links per cell. At this aggregated level, the data cannot be considered staggeringly large by any measure, quite the contrary.

Note that in the case of the information required for the sprawl indices, one needs more than 20 variables from each spatial area plus the mobility (commuting) network. The deployment of resources and massive number of interviews needed to perform a census and to gather all the land use, economic, social, etc., information to produce sprawl indices is far greater. For instance, the cost of the 2010 US census went up to 13 billion dollars. Beyond cost, the information can only be compared within a single country to ensure standardization of statistical measures. Consequently, a global comparison would be out of the question. In fact, the inherent limitations of sprawl indices and density measures have been widely discussed in:

- Ewing *et al*, Measuring sprawl and its transportation impacts. *Transportation Research Record* **1831**, 175-183 (2003).

In contrast, the trip out-flows required to calculate our metric can be obtained in real-time and at much lower cost with existing mobile technology. While we analyze information obtained from Google location history, Call Data Records and Location Based Social Networks are equally good (if not lower-resolution) sources. We already demonstrated the calculation of the flow-hierarchy from the US census, so surveys are also valid as a data source.

We have also added a new section in the Supplementary Information (**Supplementary Note 6**), where we present an analysis of Φ obtained for US cities, via three population-level models, as well as a discussion in a new section titled *Estimating the flow-hierarchy from aggregated trip-distribution models on page 7 of the manuscript*:

It is instructive to check whether the flow-hierarchy can be accurately extracted from aggregated trip distribution models such as the gravity, radiation and population-weighted opportunity [35– 37]. In principle, these models are expected to generate realistic trip flows between locations (with varying degrees of accuracy), given input information such as the population in the locations or the trip out-flows of every cell...

We demonstrate that the flow-hierarchy can also be reasonably estimated from the radiation model.

...but the actual gain in new insights / predictive power compared to more traditional and less data-intensive measures like density/sprawl index is limited.

We believe the issue of the data-intensiveness has been clarified; the flow-hierarchy is a significantly simpler, and cheaper to extract measure than the sprawl-index. Given that Φ is correlated with both the population density and sprawl (**Supplementary Figures S21, S22 and S24**), this in itself makes it a novel and useful measure to quantify the connection between mobility structure and urban indicators. Yet, as we show Φ contains much more information on urban indicators than the population density (**compare Fig. 5 with Supplementary Fig. 23**), as well as comparable (and in the case of pollution and its potential causal mechanisms) more information than the sprawl index (**see Supplementary Table 9, Supplementary Fig. 25 and Supplementary Fig. 18**). As seen from **Fig. 1c-h and Fig. 2b**, it also captures the spatial proximity of the hotspots, and indeed the population density (**Supplementary Fig. 21**). Our multivariate analysis (**Fig. 7, Supplementary Tabs. S10-S30**) indicate non-trivial information gain in a whole plethora of health and transportation indicators when taking into account Φ . Indeed, the very nature of the hotspots extracted from our data runs counter to the insight offered by the referee. Furthermore, the metric can be deployed in those places that simply don't have the resources to conduct costly

surveys to get multi-dimensional information on urban indicators. In combination, we contend that this provides exceptionally strong evidence on the novelty, efficacy, deployability and the insights gained from our analysis. We have made these points much clearer in the manuscript. On **page 8, last paragraph**:

Next we examine the connection of the flow-hierarchy with other urban indicators such as transportation, pollutant emissions, and health. We note that connections between such indicators and existing measures of urban structure such as population density and urban sprawl have been previously established and used to inform urban policy [2, 6, 16, 40–46]. The mobility patterns are ultimately shaped by urban infrastructure, population distribution and the attendant socioeconomic needs, and therefore is naturally expected to show a similar connection. Given that Φ captures the structural organization of mobility across the entire city, it is an ideal metric to study these potential relations. Indeed, cities that are more hierarchical tend to have higher population densities and are more compact (Supplementary Figs. S21, S22 and S24).

And on **page 13, last paragraph**:

While existing measures of urban structure such as population density and sprawl composite indices correlate with Φ (Supplementary Figs. 22 and 24) and with urban indicators (Supplementary Figs. 23 and 25), the flow-hierarchy conveys comparatively more information, as also demonstrated through a multivariate analysis that includes behavioral and socioeconomic factors. Furthermore, measures of urban sprawl require composite indices built up from much more detailed information on land use, population, density of jobs and street geography among others (some- times up to 20 different variables [6]). In addition to the high data requirements such metrics are also costly to obtain; censuses and surveys require a massive deployment of resources in terms of interviews, and are only standardized at a country level, hindering the correct quantification of sprawl indices at a global scale [52]. On the other hand, the flow-hierarchy, being constructed from mobility information alone, is comparatively much more accessible in terms of both cost and availability of data. Indeed, given the information content inherent in Φ —levels of hierarchy, spatial distribution of activity, quality of urban indicators—it can be deployed efficiently and at scale in those parts of the world where there is little-to-no metadata for urban indicators. The metric is completely general and non-parametric, enabling it to be extracted from any mobility data (including, as we demonstrated, commuting flows extracted from the census), although the most accurate results are obtained when the full spectrum of mobility is considered. Apart from Location History data and surveys, Φ can easily be calculated from mobile phone records [22] or Location Based Social Networks [23].

...it would perhaps be more interesting to explore the impact of the flow-hierarchy on the general mixing of people, which is more fundamental for the functioning of cities?

This is an excellent suggestion, and we have now included this analysis in the main manuscript on **page 8, second para**, as well as **Figure 4 and Supplementary Note 7, Supplementary Fig. 16**.

We consider next the relation of the mobility hierarchy with other urban characteristics. We start our analysis by examining the level of population-mixing in our considered cities. The population- mixing is a particularly relevant measure in social-science applications capturing levels of inequality and accessibility [38, 39]...

... In Fig. 4, we show the dependence of these metrics on Φ for US cities, finding a weak but monotonic and statistically significant positive trend with Φ , implying stronger levels of mixing in more hierarchical urban areas. Correlations are measured using Pearson, Spearman and LOESS with the corresponding variances denoted as $RP2$, $RS2$ and $RL2$ (Details in Supplementary Note 9). Similar results are obtained when a more involved mixing metric based on entropy measures is applied (see Supplementary Note 7 and Supplementary Fig. 16). The Gini coefficients indicate that for cities with larger Φ , the mixing is on average more concentrated due to the more intense flows of hotspots at the same level. While the stronger connections between hotspots facilitate population-mixing, we note that factors such as geography and transportation infrastructure are likely to play a role.

Indeed, we find a monotonic trend with increased hierarchy and levels of population mixing. We thank the referee for prompting us to make this analysis, as it strengthens our results.

From a policy maker's / urban planner's viewpoint, it is hard to understand how one actually could influence the flow-hierarchy.

To clarify this point, we have added the following in the discussion section in the **last paragraph of page 14**:

Given the ongoing debate on the optimal structure of cities [6, 40, 45], the flow-hierarchy, introduces a different conceptual perspective compared to existing measures, and can shed new light on the organization of cities. Indeed, while the metric is based on trip-flows, mobility, in addition to opportunity and demand, is strongly shaped by geography, land-use and quality of infrastructure. From a public-policy point of view, it is instructive to note that cities with greater degree of mobility hierarchy tend to have more desirable urban indicators. Given that this hierarchy is a measure of proximity and direct connectivity between socioeconomic hubs, a possible direction could be to shape opportunity and demand in a way that facilitates a greater degree of hub-to-hub movement than a hub-to-spoke architecture. The proximity of hubs can be generated through appropriate land use, that can be shaped by clever zoning laws in terms of business, residence or service areas. The presence of efficient public transportation and the comparatively lower use of cars is another important factor; although public transportation is not enough in terms of facilitating ease of access and lowering pollution, when countered by comparatively longer car trip-lengths as seen in the case of San Francisco. Perhaps a combination of policies, such as congestion-pricing, used to disincentivize private transportation to socioeconomic hubs, along with building public transportation in a targeted fashion to directly connect the hubs, may well prove useful.

In summary we thank the referee for the great and constructive comments, the addressal of which, we feel, has significantly improved the quality of the manuscript. We have presented exceptionally strong evidence for the novelty and use of our measure, and we look forward to a positive appraisal.

Reviewer #3:

The authors propose a novel quantitative indicator of hierarchy in urban travel. This is based on stratification of zones in urban areas according to the frequency of travel from (equivalently, to) them to indicate travel hot-spots. Cities in which a greater proportion of travel is between zones that have similar hot-spot levels (ie peer-peer) are described as having a high degree of flow hierarchy according to the authors' proposed measure (Φ), whilst cities in which more travel is between hot-spots and lesser travelled zones are described as having a low degree of flow hierarchy...The strength of the paper is that it establishes the novel Φ measure of urban mobility that is somewhat outperforms existing ones in correlation with various urban indicators of pollution and health.

We thank the referee for the positive assessment and are delighted that our analysis is considered novel and of significance.

There are several weaknesses in the evidence presented and the lines of reasoning based on them. The fundamental point for the authors to note is that the Φ measure of mobility is a descriptor of travel behaviour rather than any of urban form, infrastructure, transport provision and travel opportunity. In this sense, the novel measure is less a description of urban organisation than it is one of citizens' behaviour that represents a response to the combination of opportunity and need. Although the abstract promises relationships between urban organisation and urban indicators, what is actually provided is an observation of the relationships between citizens' travel behaviour and these indicators.

There is rather distinguished precedence in interpreting urban organization in the sense presented in this manuscript. Indeed, we followed the tradition that considers cities as a product of citizens actions as in research conducted by Michael Batty and collaborators,

- Michael Batty. *Building a science of cities*. Cities **29**, S9-S16 (2012).
- Chen Zhong, Stefan Muller Arisona, Xianfeng Huang, Michael Batty, and Gerhard Schmitt. *Detecting the dynamics of urban structure through spatial network analysis*. International Journal of Geographical Information Science **28**, 2178-2199 (2014).

and further developed in many other recent publications as for instance

- Thomas Louail, Maxime Lenormand, Oliva G Cantu Ros, Miguel Picornell, Ricardo Herranz, Enrique Frias-Martinez, Jose J Ramasco, and Marc Barthelemy. *From mobile phone data to the spatial structure of cities*. Scientific Reports **4**, 52-76 (2014).
- Minjin Lee, Hugo Barbosa, Hyejin Youn, Petter Holme, and Gourab Ghoshal. *Morphology of travel routes and the organization of cities*. Nature Communications **8**, 2229 (2017).

While mobility emerges in response to opportunity and demand, these features themselves are shaped by city infrastructure, land-use and other facets of urban organization. Admittedly, “urban organization” can also be interpreted in terms of the narrow lens of city infrastructure. Nevertheless, to avoid confusion and controversy, we modified the wording of the manuscript, in the title, the abstract as well as the main text, where in all cases we have replaced “urban organization” with the more direct “hierarchical organization of mobility”. We also make clear that the uncovered relations are between travel patterns and urban indicators on health, transport and pollution.

There are more specific weaknesses that arise from the authors' failure to consider causality. The most extreme case of this is to seek a correlation between Φ and distance to closest hospital. No explanation is offered for how either of these could affect the other

In this, we agree with the referee, our presentation was too concise, and very well may give the impression that the analysis was somewhat arbitrary. Yet there is a clear “method to the madness” as it were, and we have now made the argument much clearer. While one must display abundant caution in making statements about causality in a complex adaptive system such as a city, where plausible, we have now included both hypotheses and causal arguments for the observed relationships. As regards to proximity to hospitals, we point the referee to **last paragraph on page 11** (excerpted below) which makes the motivations and the argument clear:

We next explore the relation between Φ and health indicators on account of the strong observed relation between pollution and public wellness [47, 48]. We consider first the prevalence of ischemic stroke, whose incidence is known to be directly affected by the pollutant concentration in the atmosphere [40, 44]. While there appears weak connections between the flow-hierarchy and prevalence (Supplementary Fig. 20), we find a clear connection with the mortality rate, with a monotonically decreasing trend with increased Φ (Fig. 5g). It is well known that the survival rates for strokes is strongly dependent on timely medical attention; either provision of emergency services at home, or ease of access to hospitals [49, 50]. Indeed, Fig. 5h shows that the average distance to hospitals from any given area in the city decreases with increasing Φ . This connection is likely due to a combination of factors. First, we note the correlation between Φ and the average population density as well as its spatial distribution (Supplementary Figs. 21 and 22). Denser cities display a more hierarchical organization of mobility, with a concentration of high-level hotspots in a single area. With the reasonable assumption that medical services are distributed according to population, denser cities have comparatively higher number of such facilities in a given area as compared to sparser cities, implying more proximity to such facilities on average. Public transportation, which has a higher prevalence (on average) in more hierarchical cities, may also play a role, as it facilitates ease of travel in more dense areas as compared to using cars. However, given the comparable mortality and proximity values seen for Los Angeles and San Francisco, which differ in public transportation share, but are similar in terms of per-capita VMT, proximity to medical facilities as a function of the population density and its spatial distribution, seem to be the primary causal factors. Note that merely the average population density in itself is not sufficient to explain these trends (Supplementary Fig. 23 g-i). Indeed, the need for prompt medical attention is crucial also in traffic accidents, and we find the same connection between Φ and accident mortality in Fig. 5i even while accounting for modal share of such fatalities (Supplementary Fig. 19). The proximity of high-level hotspots in cities with large values of Φ is also the likely causal mechanism behind the observed greater pedestrian activity.

The observation that a the mode share of public transport is greater and that of car is less in cities that have a greater degree of hierarchy risks endogeneity: public transport is necessarily centralised and serves between major trip-making centres. For this reason the causality between and mode share is ephemeral.

We contend this is a hypothesis. Indeed, it is true that in cities with an extensive public transportation system, there are likely many transportation hubs, that in principle could facilitate the emergence of hotspots, enhancing the

hierarchy of mobility flows. Thus, a priori, one might expect to see a strong connection between the presence of public transportation (and its use) and stronger hierarchies in the mobility patterns. While this is true on average---and in any case, just because it seems plausible, does not mean that this effect should not be checked and quantified---the reality is far more nuanced.

We point the referee to **Fig. 5a**, where we see that Los Angeles and San Francisco have comparable levels of hierarchy, but differ noticeably in the use of cars, the modal share of public transportation and pedestrian activity. The discrepancy is even more egregious when looking at global cities as shown in **Fig. 6** (comparing Lyon with Mexico City for instance). Indeed, while the hotspots correspond to transportation centers, they are also dense residential, business and leisure areas (**Supplementary Fig. 11**). While LA and SF display large differences in the modal share of cars, the per-capita Vehicle Miles Traveled (**Supplementary Fig. 18a**) is almost identical in both cities, implying that, despite better prevalence of public transportation, residents of SF drive longer distances than those living in LA. This also has implications for some of the other concerns the referee raised on pollution, which we address in the next comment. In the meantime, we point the referee to a discussion on this feature on **page 9 last paragraph**:

We start by considering the modal transportation share. For cities with an extensive public transportation system, there are likely many transportation hubs, that in principle could facilitate the emergence of hotspots, enhancing the hierarchy of mobility flows. Thus, a priori, one would expect to see a strong connection between the presence of public transportation (and its use) and stronger hierarchies in the mobility patterns. In Figs. 5a-c, we plot Φ against various commuting modes: by car, by public transport and on foot, respectively. The Pearson R^2 , is at least 0.4, with all three p -values below 0.001 (full list of p -values for all urban indicators in Supplementary Table 6). Strongly hierarchical cities show higher levels of public transportation usage and more pedestrian trips, while transport in less hierarchical ones is dominated by the use of private cars. Similar trends exist for other transportation metrics such as vehicle-miles traveled (VMT) per capita and transit trips per capita (Supplementary Fig. 18). We note however, that while the connection between Φ and public transportation usage holds on average, the explained variance is 0.45 and there are important exceptions. In particular, both Los Angeles and San Francisco have comparable hierarchical structure, yet differ significantly in their modal share, with the latter displaying a much higher share of public transportation and lower car usage. Indeed, both Los Angeles and San Francisco have comparable values of per-capita VMT (Supplementary Figure 18a) despite the differences in car modal share, indicating that residents of San Francisco typically travel longer distances than those in Los Angeles. This implies that a centralized public transportation network by itself does not facilitate hierarchy in mobility flows, instead other factors such as spatial constraints, geographic impediments and corresponding land usage are important factors.

The observation that emissions of pollutants tend to be lower in cities that have high values is clearly exposed to common causation.

Once again, the situation is far more nuanced, and this is a valid but ultimately not completely substantiated hypothesis. Yes, our decision to perform this analysis was motivated by the observed connections with modal share. However, beyond the share of private vehicle usage, factors contributing to pollution include quality and extent of infrastructure, land use distribution, geographical features, congestion, etc. Once again using LA and SF as archetypal examples, they differ in transportation patterns, but are very much alike in pollution indicators (**Fig. 5d-f**). Furthermore, the functional relationship between Φ and the modal share is quite different from its connection with pollution indicators (even though the trend is monotonic). It appears that the observed emission trends are better explained by the per-capita Vehicle Miles Traveled (Supplementary Figure 18a), which as mentioned earlier, is comparable in LA and SF despite their considerable differences in modal share. The functional dependence between Φ and VMT is also more akin to that seen in its relationship with pollution indicators. So we cannot agree with the reviewer's comments "(it is) clearly exposed to common causation". To expound upon this, we have added the following, starting on the last para of page 10:

Given the established connection between car use and pollution [45, 46], one might expect to see a similar relation between Φ and pollution indicators. Yet, once again, the emission of pollutants is not only related to the use of transportation modes. The quality of transportation infrastructure, congestion and geographical features also play an important role. In Figs. 5d-f, we plot Φ against emission of NO_x , CO and Particulate Matter (PM10), finding that pollution emissions are anti-correlated with higher levels of hierarchy (See Supplementary Table 7 for connection with $\text{PM}_{2.5}$ emissions). However, the shape of the LOESS fit, and the general trend, in Fig. 5a and d-f

are very different. Indeed, it appears that the total emissions are shaped more by the per-capita VMT than the car modal share, with similar shapes for the LOESS fit. This is confirmed by the fact that Los Angeles and San Francisco are quite comparable in their pollution emissions and VMT values, despite the large differences in modal share. Given that Φ also captures the spatial distribution of the hotspots—the distance between which influences the length of trips—if hotspots are organized as in Paris (Fig. 1c) with a single clear nucleus, as opposed to the more scattered configuration of Los Angeles (Fig. 1e), Φ is higher and the trips tend to be shorter. Of course, factors such as congestion also play a role.

The present manuscript is observational in nature and lacks hypothesis.

We believe that in our preceding comments, we have addressed these concerns. We note that in complex adaptive systems such as cities, the interactions between components is highly non-linear, multivariate, and consisting of various feedback loops. In such a setting, making strong claims about causal mechanisms is problematic. Nevertheless, where plausible, and where we have compelling evidence, we have augmented our findings with causal explanations. Furthermore, when presenting our findings, in each case we have included a hypothesis, similar to the line of argumentation proposed by the referee. Of course, our findings contradict some of the straightforward hypotheses, thus making our results, we believe, more intriguing.

Additionally, we would like to stress that the leitmotiv of this work is the introduction of a new metric able to characterize urban mobility, and revealing through it, new insights on the dynamics in cities and its relation with urban indicators. In this sense, Φ should be considered in the same context as population density and sprawl indices (as studied extensively in Refs. [2,6,16,40-46,52] of the main manuscript). Indeed, in all of these prior publications, attempts were made to connect the urban indicators studied in this manuscript with other metrics of urban organization. Given that we propose the flow-hierarchy as a superior metric to capture such aspects of urban systems, it is natural that we should conduct such an analysis.

The flow-hierarchy is a significantly simpler, and cheaper to extract measure than the sprawl-index. Given that Φ is correlated with both the population density and sprawl (**Supplementary Figures S21, S22 and S24**), this in itself makes it a novel and useful measure to quantify the connection between mobility structure and urban indicators. Yet, as we show, Φ contains much more information on urban indicators than the population density (**compare Fig. 5 with Supplementary Fig. 23**), as well as comparable (and in the case of pollution and its potential causal mechanisms) more information than the sprawl index (**see Supplementary Table 9, Supplementary Fig. 25 and Supplementary Fig. 18**). As seen from **Fig. 1c-h and Fig. 2b**, it also captures the spatial proximity of the hotspots, and indeed the population density (**Supplementary Fig. 21**). Our multivariate analysis (**Fig. 7, Supplementary Tabs. S10-S30**) indicate non-trivial information gain in a whole plethora of health and transportation indicators when taking into account Φ . Furthermore, the metric can be deployed in places simply it does not exist the resources to conduct costly surveys to get multi-dimensional information on urban indicators. In combination, we contend that this provides exceptionally strong evidence on the novelty, efficacy, deployability and the insights gained from our analysis. We have made these points much clearer in the manuscript. On **page 13, last para**:

While existing measures of urban structure such as population density and sprawl composite indices correlate with Φ (Supplementary Figs. 22 and 24) and with urban indicators (Supplementary Figs. 23 and 25), the flow-hierarchy conveys comparatively more information, as also demonstrated through a multivariate analysis that includes behavioral and socioeconomic factors. Furthermore, measures of urban sprawl require composite indices built up from much more detailed information on land use, population, density of jobs and street geography among others (some- times up to 20 different variables [6]). In addition to the high data requirements such metrics are also costly to obtain; censuses and surveys require a massive deployment of resources in terms of interviews, and are only standardized at a country level, hindering the correct quantification of sprawl indices at a global scale [52]. On the other hand, the flow-hierarchy, being constructed from mobility information alone, is comparatively much more accessible in terms of both cost and availability of data. Indeed, given the information content inherent in Φ —levels of hierarchy, spatial distribution of activity, quality of urban indicators—it can be deployed efficiently and at scale in those parts of the world where there is little-to-no metadata for urban indicators. The metric is completely general and non-parametric, enabling it to be extracted from any mobility data (including, as we demonstrated, commuting flows extracted from the census), although the most accurate results are obtained when

the full spectrum of mobility is considered. Apart from Location History data and surveys, Φ can easily be calculated from mobile phone records [22] or Location Based Social Networks [23].

The findings are descriptive and offer no guidance on policy.

We have now included a description on potential policy prescriptions in the discussion section in the **last para of page 14**:

Given the ongoing debate on the optimal structure of cities [6, 40, 45], the flow-hierarchy, introduces a different conceptual perspective compared to existing measures, and can shed new light on the organization of cities. Indeed, while the metric is based on trip-flows, mobility, in addition to opportunity and demand, is strongly shaped by geography, land-use and quality of infrastructure. From a public-policy point of view, it is instructive to note that cities with greater degree of mobility hierarchy tend to have more desirable urban indicators. Given that this hierarchy is a measure of proximity and direct connectivity between socioeconomic hubs, a possible direction could be to shape opportunity and demand in a way that facilitates a greater degree of hub-to-hub movement than a hub-to-spoke architecture. The proximity of hubs can be generated through appropriate land use, that can be shaped by clever zoning laws in terms of business, residence or service areas. The presence of efficient public transportation and the comparatively lower use of cars is another important factor; although public transportation is not enough in terms of facilitating ease of access and lowering pollution, when countered by comparatively longer car trip-lengths as seen in the case of San Francisco. Perhaps a combination of policies, such as congestion-pricing, used to disincentivize private transportation to socioeconomic hubs, along with building public transportation in a targeted fashion to directly connect the hubs, may well prove useful.

Beyond this, there are several areas where the text requires improvement. These include, but are not necessarily limited to:

- o confusion in the text between "significant" and various of {large, statistical, important, substantial}

Addressed where applicable.

- o the term "population flow" is unusual in the transport field: the population itself does not relocate. The terms "travel" or "travel behaviour" might be used instead, at the authors' discretion (and responsibility)

We changed this to trip-flows.

- o The penultimate sentence ("Specifically, we show ...hierarchy." of the introductory section is opaque and requires clarification.

The discussion has been changed substantially

- o The privacy-protecting noise mechanism described in the section "Aggregation of Trip flows" - specifically "This automated mechanism ... very strong guarantee." conveys little information on the processes followed and should therefore be clarified.

We have included much more details on this, starting from the last paragraph on page 2.

- o The symbol Φ is absent from captions to Fig 2 and Fig 4 so should be inserted

Done.

- o The caption to Fig 2 refers erroneously to Fig 2c-h (perhaps Fig 1c-h is intended?)

Corrected, thank you.

- o The calculation on page 5 of $\Phi_u = 0.21$ is not consistent with the formula and parameter values specified.

Thank you, the correct value is 0.204. We erroneously rounded up. It has now been corrected to 0.20

In summary, a sincere vote of thanks to the referee for prompting us to dig deeper into our findings. We hope we have addressed all of the stated concerns, which in our opinion has improved substantially the readability and reach of the manuscript. We look forward to, and are hopeful, of a positive appraisal.

REVIEWERS' COMMENTS:

Reviewer #1 (Remarks to the Author):

I don't have any further comment.

Reviewer #2 (Remarks to the Author):

At this stage, I have the following two recommendations:

1. I appreciate the additional analysis of the population mixing. The correlations between ϕ and the population mixing measure (new Fig. 4) is however weak and the possible explanation offered is in my view not fully convincing. I believe one would really need to quantify the co-location events between individuals (or population groups) to derive a more solid conclusion. I recommend moving this part into the supplementary material.
2. Because the authors decided to stay with the use a single method to characterize the hierarchical structure of the mobility flows, I recommend to at least address the potential impact of changing the spatial resolution on the conclusions of this paper, especially as most correlations are not very strong and lack in-depth discussion. This issue of spatial granularity has perhaps already been discussed in previous work introducing the Loubar method, but addressing it again would increase trust in the method.

Reviewer #1

I don't have any further comment.

We thank the reviewer for the comments and constructive feedback.

Reviewer #2

At this stage, I have the following two recommendations:

- 1. I appreciate the additional analysis of the population mixing. The correlations between ϕ and the population mixing measure (new Fig. 4) is however weak and the possible explanation offered is in my view not fully convincing. I believe one would really need to quantify the co-location events between individuals (or population groups) to derive a more solid conclusion. I recommend moving this part into the supplementary material.*

We agree, more information on trajectories would be needed to properly quantify the level of mixing beyond the origin and destination of the trips. We have followed the reviewer's suggestion and moved the analysis to the SI.

- 2. Because the authors decided to stay with the use a single method to characterize the hierarchical structure of the mobility flows, I recommend to at least address the potential impact of changing the spatial resolution on the conclusions of this paper, especially as most correlations are not very strong and lack in-depth discussion. This issue of spatial granularity has perhaps already been discussed in previous work introducing the Loubar method, but addressing it again would increase trust in the method.*

We agree that this is an important point and have added new material in Supplementary Note 4 with a study of the effect of spatial scale on the flow-hierarchy (Supplementary Fig. 9). While, it increases slowly as one approaches cells corresponding to the size of the city, the relative ranking of the studied cities, and, therefore, the correlations with urban indicators is quite stable and robust across multiple scales.

In summary, we thank the reviewer for the very constructive comments, leading to a significant improvement in the quality of the manuscript.